# Unveiling COVID-19 from CHEST X-Ray with Deep Learning: A Hurdles Race with Small Data

**DOI:** 10.3390/ijerph17186933

**Published:** 2020-09-22

**Authors:** Enzo Tartaglione, Carlo Alberto Barbano, Claudio Berzovini, Marco Calandri, Marco Grangetto

**Affiliations:** 1Computer Science Department, University of Turin, 10149 Torino, Italy; carlo.barbano@edu.unito.it (C.A.B.); marco.grangetto@unito.it (M.G.); 2Azienda Ospedaliera Città della Salute e della Scienza Presidio Molinette, 10126 Torino, Italy; cberzovini@cittadellasalute.to.it; 3Oncology Department, University of Turin, AOU San Luigi Gonzaga, 10043 Orbassano, Italy; marco.calandri@unito.it

**Keywords:** chest X-ray, deep learning, classification, COVID-19

## Abstract

The possibility to use widespread and simple chest X-ray (CXR) imaging for early screening of COVID-19 patients is attracting much interest from both the clinical and the AI community. In this study we provide insights and also raise warnings on what is reasonable to expect by applying deep learning to COVID classification of CXR images. We provide a methodological guide and critical reading of an extensive set of statistical results that can be obtained using currently available datasets. In particular, we take the challenge posed by current small size COVID data and show how significant can be the bias introduced by transfer-learning using larger public non-COVID CXR datasets. We also contribute by providing results on a medium size COVID CXR dataset, just collected by one of the major emergency hospitals in Northern Italy during the peak of the COVID pandemic. These novel data allow us to contribute to validate the generalization capacity of preliminary results circulating in the scientific community. Our conclusions shed some light into the possibility to effectively discriminate COVID using CXR.

## 1. Introduction

COVID-19 virus has rapidly spread in mainland China and into multiple countries worldwide [1]. As of 9 August 2020, 19,432,244 Patients with COVID-19 have been recorded, and 721,594 of them died [2].

Early diagnosis is a key element for proper treatment of the patients and prevention of the spread of the disease. Given the high tropism of COVID-19 for respiratory airways and lung epythelium, identification of lung involvement in infected patients can be relevant for treatment and monitoring of the disease.

Virus testing is currently considered the only specific method of diagnosis. The Center for Disease Control (CDC) in the US recommends collecting and testing specimens from the upper respiratory tract (nasopharyngeal and oropharyngeal swabs) or from the lower respiratory tract when available (bronchoalveolar lavage, BAL) for viral testing with reverse transcription polymerase chain reaction (RT-PCR) assay [3]. Testing on BAL samples provides higher accuracy, however this test is unconfortable for the patient, possibly dangerous for the operator due to aerosol emission during the procedure and cannot be performed routinely. Nasopharingeal swabs are instead easily executable and affortable and current standard in diagnostic setting; their accuracy in literature is influenced by the severity of the disease and the time from symptoms onset and is reported up to 73.3% [4].

Current position papers from radiological societies (Fleischner Society, SIRM, RSNA) [3,5,6] do not recommend routine use of imaging for COVID-19 diagnosis.

However, it has been widely demonstrated that, even at early stages of the disease, chest x-rays (CXR) and computed tomography (CT) scans can show pathological findings. It should be noted that they are actually non specific, and overlap with other viral infections (such as influenza, H1N1, SARS and MERS): most authors report peripheral bilateral ill-defined and ground-glass opacities, mainly involving the lower lobes, progressively increasing in extension as disease becomes more severe and leading to diffuse parenchymal consolidation [7,8]. CT is a sensitive tool for early detection of peripheral ground glass opacities; however routine role of CT imaging in these Patients is logistically challenging in terms of safety for health professionals and other patients, and can overwhelm available resources [9].

Chest X-ray can be a useful tool, especially in emergency settings: it can help exclude other possible lung “noxa”, allow a first rough evaluation of the extent of lung involvement and most importantly can be obtained at patient’s bed using portable devices, limiting possible exposure in health care workers and other patients. Furthermore, CXR can be repeated over time to monitor the evolution of lung disease [5].

Wong et al. in a study recently published on Radiology reported that x-ray has a sensitivity of 69% and that the severity of CXR findings peaked at 10–12 days from the date of symptom onset [8].

Because of their mostly peripheral distribution, subtle early findings on CXRs may be a diagnostic challenge even for an experienced thoracic radiologist: in fact, there are many factors that should be taken into account in image interpretation and that could alter diagnostic performance (such as patient body type, compliance in breath-holding and positioning, type of projection that can be executed i.e., antero-posterior in more critical patients examined at bedside, postero-anterior if the patient can be moved to radiology unit and is more collaborating, presence of other medical devices on the thorax, especially in x rays performed in intensive care units, etc.). In the challenging and never-before seen scenario that rose to attention in the last months, radiologists may look at Artificial Intelligence and deep learning applications as a possible aid for daily activity, in particular for identification of the more subtle findings that could “escape” the human eye (i.e., reduce false-negative x-rays) or, on the other side, could prompt swab repetition of further diagnostic examinations when first virus testing is negative (considering its sub-optimal sensitivity).

Given the intrinsic limits of CXR but at the same time its potential relevant role in the fight against COVID-19, in this work we set up a state of the art deep learning pipeline to investigate if computer vision can unveil some COVID fingerprints. It is evident that the answer will be given only when publicly available large image datasets empower scientists to train complex neural models, to provide reproducible and statistically solid results and to contribute to the clinical discussion. Unfortunately, up to date, we are stuck with few labelled images. Thanks to the collaboration with the radiology unit of Città della Salute e della Scienza di Torino (CDSS) hospital in Turin in the last days of March (at the peak of epidemic in Italy), we managed to collect the COVID Radiographic images Data-set for AI (CORDA), currently comprising images from 386 Patients that underwent COVID screening.

## 2. Materials and Methods

It is evident that currently there is not yet a significant amount of work devoted to automatic detection of COVID from medical imaging. Nonetheless, one can refer to previous epidemics caused by novel strain of coronavirus such as severe acute respiratory syndrome (SARS), first recognized in Canada in March 2003, characterised by similar lung condition, i.e., interstitial pneumonia [10]. Most results leverage on the use of high resolution CT scans. As an example, in [11] CNN are investigated for classification of interstitial lung disease (ILD). Also [12,13] show that deep learning can be used to detect and classify ILD tissue. The authors of [13] focus on a design a CNN tailored to match the ILD CT texture features, e.g., small filters and no pooling to guarantee spatial locality.

Fewer contributions focus on classification of X-ray chest images to help SARS diagnosis: in [14] lung segmentation, followed by feature extraction and three classification algorithms, namely decision tree, shallow neural network and classification and regression tree are compared, the latter yielding the higher accuracy on the SARS detection task. However, on the pneumonia classification task, NN-based approaches show encouraging results. In [15] texture features for SARS identification in radiographic images are proposed and designed using signal processing tools.

In the last days a number of pre-prints targeting COVID classification with CNN on radiographic images have begun to circulate thanks to open access archives. Many approaches have been taken to tackle the problem of classifying chest X-ray scans to discriminate COVID-positive cases. For example, Sethy et al. compare classification performances obtained between some of the most famous convolutional architectures [16]. In particular, they use a transfer learning-based approach: they take pre-trained deep networks and they use these models to extract features from images. Then, they train a SVM on these “deep features” to the COVID classification task. A similar approach is also used by Apostopolous et al.: they pre-train a neural network on a similar task, and then they use the trained convolutional filters to extract features, on top of which a classifier attempts to select COVID features [17].

Narin et al. make use of resnet-based architectures and the recent Inception v3 and then they use a 5-fold cross validation strategy [18]. Finally, Wang et al. propose a new neural network architecture to be trained on the COVID classification task [19].

All of these approaches use a very small dataset, *COVID-ChestXRay* [20], consisting of approximately 100 COVID cases considering CXR only, at the time of writing. Furthermore, in order to build COVID negative cases, typically data are sampled from other datasets (mostly, from *ChestXRay*). However, this introduces a potential issue: if any bias is present in the dataset (a label in the corners, a medical device, or other contingent factors like similar age, same sex etc.) the deep model could learn to recognize these dataset biases, instead of focusing on COVID-related features.

These works present some potential issues to be investigated:Transfer learning: in the literature it is widely recognized that transfer learning-based approaches prove to be effective, also for medical imaging [21]. However, it is very important to be careful on the particular task the feature extractor is trained on: if such task is very specific, or contains biases, then the transfer learning approach should be carefully carried on.Hidden biases in the dataset: most of the current works rely on very small datasets, due to the limited availability of public data on COVID positive cases. These few data, then, contain little or even no metadata on age, gender, different pathologies also present in these subjects, and other necessary information necessary to spot on this kind of biases. Besides these, there are other biases we can try to correct. For example, every CXR has its own image windowing parameters or other acquisition settings that a deep model could potentially learn to discriminate. For example, one model may cluster images according to the scan tool used for the exam; if some scan settings correspond to all COVID examples, these generate a spurious correlation that the model can exploit to yield apparently optimal classification accuracy. Another example is given by textual labeling in images: if all the negative examples are sampled from the same dataset, the deep model could learn to recognize such feature instead of focusing on the lung content etc.Very small test sets: as a further consequence of having very little data, test set sizes are extremely small and they do not provide any statistical certainty on learning.

In this work we do not mean to answer whether and how CXR can be used in the early diagnosis of COVID, but to provide a methodological guide and critical reading of the statistical results that can be obtained using currently available datasets and learning mechanisms. Our main contribution is an extensive experimental evaluation of different combinations of usage of existing datasets for pre-training and transfer learning of standard CNN models. Such analysis allows us to raise some warnings on how to build datasets, pre-process data and train deep models for COVID classification of X-ray images. We show that, given the fact that datasets are still small and geographically local, subtle biases in the pre-trained models used for transfer learning can emerge, dramatically impacting on the significance of the performance one achieves.

## 3. Results

In this section we are going to describe the proposed deep-learning approach based on quite standard pipeline, namely chest image pre-processing and lung segmentation followed by classification model obtained with transfer learning. Data pre-processing is fundamental to remove any bias present in the data: we will show that it is easy for a deep model to recognize these biases which drive the learning process. Given the small size of COVID datasets, a key role is played by the larger datasets used for pre-training. Therefore, we first discuss which datasets can be used for our goals.

### 3.1. Datasets

For the experiments we are going to show, six different datasets are used. Four of these datasets provide a label for COVID classification task (*COVID-ChestXRay, CORDA, ChestXRay* and *RSNA*) while the other two (*Montgomery County X-ray Set* and *Shenzhen Hospital X-ray Set*) provide a segmentation mask for lungs; these two are used in the pre-processing phase only. In the following we briefly recall the main characteristics of each dataset:*COVID-ChestXRay*: this dataset was developed by gathering CXR and CT images from various website and publications. At the time of writing, it comprises 287 images with different type of pneumonia (COVID-19, SARS, MERS, Streptococcus spp., Pneumocystis spp., ARDS) [20]. Currently, a subset of 137 CXRs (PA) containing 108 COVID positive images and 29 COVID negatives is available at https://github.com/ieee8023/covid-chestxray-dataset.*CORDA*: this dataset was created for this study by retrospectively selecting chest x-rays performed at a dedicated Radiology Unit in a reference Hospital in Piedmont (CDSS) in all patients with fever or respiratory symptoms (cough, shortness of breath, dyspnea) that underwent nasopharingeal swab to rule out COVID-19 infection. Patients were collected over a 15-day period between the 16th and 30th March, 2020. It contains 447 CXRs from 386 patients, with 150 images coming from COVID-negative patients and 297 from positive ones. Patients’ average age is 61 years (range 17–97 years old). The data collection is still in progress, with other 5 hospitals in Italy willing to contribute at time of writing. We plan to make CORDA available for research purposes according to EU regulations as soon as possible.*ChestXRay*: this dataset contains 5857 X-ray images collected at the Guangzhou Women and Children’s Medical Center, Guangzhou, China. In this dataset, three different labels are provided: normal patients (1583), patients affected by bacterial pneumonia (2780) and affected by viral pneumonia (1493). This dataset was collected before COVID pandemic and is granted under CC by 4.0 and is part of a work on Optical Coherence Tomography [22]. The dataset is publicly available at https://data.mendeley.com/datasets/rscbjbr9sj/2/files/f12eaf6d-6023-432f-acc9-80c9d7393433.*RSNA*: developed by the joint effort of the *Radiological Society of North America*, *US National Institute of Health*, *The Society of Thoracic Radiology* and *MD.ai* for the RSNA Pneumonia Detection Challenge, this dataset contains pneumonia cases found in the NIH Chest X-ray dataset [23]. It comprises 20,672 normal CXR scans and 6012 pneumonia cases, for a total of 26,684 images. The dataset is publicly available at https://www.kaggle.com/c/rsna-pneumonia-detection-challenge. As the previous one, this dataset was created before COVID pandemic and therefore those not report COVID positive cases.*Montgomery County X-ray Set*: the X-ray images in this dataset have been acquired under a tuberculosis control program of the Department of Health and Human Services of the Montgomery County, MD, USA. Such a dataset contains 138 samples: 80 are normal patients and 58 are abnormal. In these images lungs have been manually segmented. The dataset is open-source and available at http://openi.nlm.nih.gov/imgs/collections/NLM-MontgomeryCXRSet.zip.*Shenzhen Hospital X-ray Set*: the X-ray images in this dataset have been collected by Shenzhen No.3 Hospital in Shenzhen, Guangdong providence, China. This dataset contains a total of 662 images: 326 images are from healthy patients while 336 show abnormalities. Such a dataset is also open-source and available at http://openi.nlm.nih.gov/imgs/collections/ChinaSet_AllFiles.zip. Ground truths for this dataset have been provided by Stirenko et al. [24].

### 3.2. Pre-Processing

For our simulations we propose a pre-processing strategy aiming at removing bias in the data. This step is very important in a setting in which we train to discriminate different classes belonging to different datasets: a neural network-based model might learn the distinction between the different dataset biases and from them “learn” the classification task. The proposed pre-processing chain is summarized in Figure 1 and is based on the following steps:Histogram equalization: when acquiring a CXR, the so-called radiographic contrast depends on a large variety of factors, typically depending on subject contrast, receptor contrast or other factors like scatter radiations [25]. Hence, the raw acquisition has to be filtered through Value Of Interest transformation. However, due to different calibrations, different range dynamics can be covered, and this potentially is a bias. Histogram equalization is a simple mean to guarantee quite uniform image dynamic in the data.Lung segmentation: the lung segmentation problem has been already faced and successfully tackled [26,27,28]. Being able to segment the lungs only, discarding all the rest of the CXRs, potentially prunes away possible bias sources, like for example the presence of medical devices (typically correlated to sick patients), various text which might be embed in the scan etc. In order to address this task, we train a U-Net [29] on *Montgomery County X-ray Set* and *Shenzhen Hospital X-ray Set*. The lung masks obtained are then blurred to avoid sharp edges using a 3 pixel radius. An example of the segmentation outcome is shown in Figure 2.Image intensity normalization in the range [0, 1].

### 3.3. Training

After data have been pre-processed, a deep model is trained. Towards this end, the following choices have been taken:Pre-training the feature extractor (i.e., the convolutional layers of the CNN). In particular, the pre-training is performed on a related task, like pneumonia classification for CXRs. It has been shown that such an approach can be effective for medical imaging [11], in particular when the amount of available data is limited as in our classification task. Clearly, pre-training the feature extractor on a larger dataset containing related features may allow us to exploit deeper models, potentially exploiting richer image feature.The feature extractor is then fine-tuned on COVID data. Freezing it certainly prevents over-fitting the small COVID data; however, we have no warranty that COVID related features can be extracted at the output of a feature extractor trained on a similar task. Of course, its initialization on a similar task helps in the training process, but in any case a fine-tuning is still necessary [30].Proper sizing of the encoder to-be-used is an issue to be addressed. Despite many recent works use deeper architectures to extract features on the COVID classification task, larger models are prone to over-fit data. Considering the minimal amount of data available, the choice of the appropriate deep network complexity significantly affects the performance.Balancing the training data is yet another extremely important issue to be considered. Unbalanced data favor biases in the learning process [31] and the choice of the data to include in the learning process is critical.Data augmentation techniques should be carefully used in such context. No generic plastic deformations for the CXR images can be safely introduced since the basic lung structure is typically the same for any human subject, and should be consistently realistic through all the augmented samples. Towards this end, rigid transformations (translation, rotation) are the only data augmentation transformations safely applicable in such context.Testing with different data than those used at training time is also fundamental. Excluding from the test-set exams taken from patients already present in the training-set is important to correctly evaluate the performance and to exclude the deep model has not learned a “patient’s lung shape” feature.Of course many other issues have to be taken into account at training time, like the use of a validation-set to tune the hyper-parameters, using a good regularization policy etc. but these very general issues have been exhaustively discussed in many other works [32,33,34].

An overall summary of pre-training, training and testing is summarized in Figure 3. The experiments discussed in the following have been designed to investigate three key aspects:Pre-training of the feature extractor: the feature extractor can be pre-trained on large generic CXR datasets, or can not be pre-trained.Composition of the training-set: the CORDA dataset is unbalanced (in fact, there is a prevalence of positive COVID cases) and some data balancing is possible, borrowing samples from publicly available non-COVID datasets. A summary of the dataset composition is displayed in Table 1. For all the datasets we used 70% of data at training time and 30% as test-set. Training data are then further divided in training-set (80%) and validation-set (20%). Training-set data are finally balanced between COVID+ and COVID—: where possible, we increased the COVID—cases (CORDA&ChestXRay, CORDA&RSNA), where not possible we sub-sampled the more populated class. This percentages were not used for the COVID-ChestXRay dataset: in this case only 15 samples are used for testing in order to compare with other works [16,17,18] that use the same partitioning. Please notice that, through all the datasets, test data are mutually exclusive with training ones, and are never used at training time.Testing on different datasets: in order to observe the possible presence of hidden biases, testing on different, qualitatively-similar datasets is a necessary step.

A summary of the most salient experimental results obtained on a combination of different datasets (Table 1) is reported in Table 2. The complete results are reported in Appendix A. All the simulations have been run on a Tesla T4 GPU using PyTorch 1.4. The source code is available at https://github.com/EIDOSlab/unveiling-covid19-from-cxr.

For all of the trained models, a number of metrics [35] have been evaluated:AUC (area under the ROC curve), provides an aggregate measure of performance across all possible classification thresholds. For every other metric, the classification threshold is set to 0.5;sensitivity;specificity;BA (balanced accuracy), since the test-set might be un-balanced;DOR (diagnostic odds ratio).

In Table 2 we compare four alternative neural network architectures, i.e., ResNet-18 [36], Resnet-50 [36], COVID-Net [19] and DenseNet-121 [37] with different combinations of datasets used for pre-training, training and testing, respectively (see columns 2–4 where datasets are identified according to labels in Table 1). As many work in the literature [17,19], we observe that, using the same COVID dataset as source for training and testing images, the performance of the deep learning models looks amazingly good. As an example, DenseNet-121 trained on images from COVID-ChestXRay dataset (D) yields BA as high as 0.9 when testing is done on corresponding testing set D. However, when testing on extra data still belonging to the same domain (they are still CXR images which undergo the same pre-processing as the training images), the performance drops significantly. In the DenseNet-121 case, we report BA of 0.53 when the same model is tested with images from CORDA datasets (A).

## 4. Discussion

In this section we analyze the results found in Section 3.3. Considering the complexity and the importance of the considered topic, we divide our analysis into three main aspects:impact of pre-training for COVID detection (Section 4.1) and how should it be performed (Section 4.2);effect of augmenting the COVID datasets with negative cases (Section 4.3);selection of the proper architecture for the COVID detection (Section 4.4 and Section 4.5).

### 4.1. To Pre-Train or Not to Pre-Train?

One very important issue to pay attention to is whether to pre-train the feature extractor or not. Given the large availability of public data for pneumonia classification (for example, in this scope we used ChestXRay and RSNA), it could be a good move to pre-train the encoder, and effectively this is what we observe looking at Table 2. For example, if we focus on the results obtained training on the CORDA dataset, without a pre-trained encoder, BA and DOR are lower than pre-training with ChestXRay or RSNA. Despite the sensitivity remains very similar, pre-training the encoder helps in improving the specificity: on the test-set extracted from CORDA, using a pre-trained encoder on RSNA, the specificity is 0.80, while it is only 0.58 with no pre-trained feature extractor. Similar improvements in the specificity can be observed also on test-sets extracted from all the other datasets, except for ChestXRay. In general, a similar behavior can be observed when comparing results for differently pre-trained encoders trained on the same dataset.

Pre-training is important; however, we can not just “freeze” the encoder on the pre-trained values. Since the encoder is pre-trained on a similar, but different task, there is no warranty the desired output features are optimal for the given classification task, and a fine-tuning step is typically required [38].

### 4.2. Pre-Training on Different Datasets

Focusing on pre-trained encoders, we show results for encoders pre-trained on two different datasets: ChestXRay and RSNA. While RSNA is a more generic pneumonia-segmentation dataset, ChestXRay contains information also about the type of pneumonia (bacterial or viral); so, at a first glance it looks a better fit for the pre-training. However, if we look at training on the CORDA dataset, we see that for the same sensitivity value, we get typically higher specificity scores for RSNA pre-training. This is not the same we observe when we compare results on the publicly-available COVID-ChestXRay: in this case, sensitivity and specificity are higher when we pre-train on ChestXRay. Looking at the same pre-trained encoder, let us say ChestXRay, we can compare results training on CORDA and on COVID-ChestXRay, which are the two COVID datasets: CORDA shows a lower sensitivity, but in general a higher specificity, except for the ChestXRay dataset. Having very little data at training time, pre-training introduces some priors in the choice of the features to be used, and depending on the final classification task, performance changes, yielding very good metric in some cases. Pre-training on more general datasets, like RSNA, in general looks a slightly better choice than using a more specific dataset like ChestXRay.

### 4.3. Augmenting COVID—Data with Different Datasets

For each and every simulation, performance on different test-sets is evaluated. This gives us hints on possible biases introduced by different datasets used at training time.

A general trend can be observed for many COVID—augmented training-sets: the BA and DOR scores measured on the test-set built from the same dataset used at training time are typically very high. Let us focus on the ChestXRay pre-trained encoder. When we train on CORDA&ChestXRay, the BA score measured on the test-set from the same dataset is 0.9 and the DOR is 122.67. However, its generalization capability for a different composition of the test-set, let us say, CORDA&RSNA, is way lower: the BA is 0.56 and the DOR 2.26 only. The same scenario can be observed when we train on CORDA&RSNA: on its test-set the BA is 0.90 and DOR 122.64, while on the test-set of CORDA&ChestXRay the BA is 0.59 and DOR 2.47. The key to understand these results lies again in the specificity score: this score is extremely high for the test-set of the same dataset the training is performed on (for example, for CORDA&RSNA is 0.95 and for CORDA&ChestXRay is 0.94) while for the others is extremely low. Such a behavior is due to the presence of some common features in all the data belonging to the same augmenting dataset. This can be observed, for example, in Figure 4a, where the extracted features from an encoder pre-trained on ChestXRay and trained on CORDA&ChestXRay are clustered using t-distributed stochastic neighbor embedding (t-SNE) [39] (blue and orange dots represent ChestXray and CORDA data samples respectively, regardless of the COVID label). T-SNE is a popular nonlinear dimensionality reduction algorithm which models each high-dimensional object to a low-dimensional point for visualization purposes: the more a low-dimensional point is far from another, the more the two objects are different. It can be noted that CORDA samples, regardless the COVID+ or COVID—label, are clearly separable from ChestXRay data. Of course, all ChestXRay images have COVID—label, so someone could argue that the COVID feature has been captured. Unfortunately we have a counterexample: in Figure 4b we compare CORDA vs. RSNA samples, using the same ChestXRay pre-trained encoder and now RSNA and CORDA samples no longer form clear clusters. Hence, the deep model specializes not in recognizing COVID features, but in learning the common features in the same dataset used at training time. We would like to remark that for all the data used at training or at test time, all the pre-processing presented in Section 3.2 has been used. We ran the same experiments without that pre-processing and performance on different datasets than the one used at training time gets even worse. For example, pre-training and training on CORDA&ChestXRay without pre-processing lowers the BA to 0.73 and the DOR to 8.31 on CORDA, while from Table 2 we have higher scores on the test set (BA of 0.91 and DOR of 122.67).

Dealing with generality of the results is a very delicate matter: what it is possible to see in Table 2 is that augmenting data with COVID—data needs to be very thoughtful since the classification performance may vary from very high accuracy down to almost useless discriminative power. Nonetheless, training using only COVID datasets yields some promising scores: for example, using ChestXRay pre-trained encoder and CORDA for training and testing, the BA we achieve is 0.56 and the DOR is 1.64. Including also COVID-ChestXRay for training (which consists in having more COVID+ and COVID—examples) improves the BA to 0.62 and the DOR to 2.93. In this case, however, the specificity is an issue, since we lack of COVID—data. However, these results show some promise that can be confirmed only by collecting large amount of data in the next months.

### 4.4. How Deep Should We Go?

After reviewing results on ResNet-18, we move to similar experiments run on the deeper ResNet-50 and DenseNet-121 shown in Table 2. The hope is that a deeper network could extract more representative features for the classification task. Given the discussion in Section 4.1, we show only the cases with pre-training of the feature extractor. Using this deeper architecture, we can observe that all the discussions made for ResNet-18 still holds. In some cases performance impairs slightly: for example, the DOR score on CORDA&ChestXRay for ResNet-18 was 122.67 while for ResNet-50 and DenseNet-121 drops to 73.35 and 86.56 respectively. This is a sign of over-fitting: given the very small quantity of data currently available, using a small convolutional neural network is sufficient and safer. Taking an opposite approach, we tried to use a smaller artificial neural network, made of 8 convolutional layers and a final fully-connected layer, which takes inspiration from the ALL-CNN-C architecture [40]. We call this architecture “Conv8”. The results on this smaller architecture are similar to those observed in Table 2. For example, training the model on CORDA dataset, on Conv8 we have a BA of 0.61 and DOR of 2.38 while for ResNet-18 with encoder pre-trained on RSNA we have BA of 0.67 and DOR 4.78. We can conclude that using a smaller architecture than ResNet-18 does not give relevant training advantages, while by using larger architectures we might over-fit data.

### 4.5. Comparison between Deep Networks Trained on Covid-Chestxray

All the observations on train and test data made above are also valid for the recently published results on the COVID classification from CXR [16,17,18,19]. One very promising approach is COVID-Net [19]. They also share the source code and the trained model, available at https://github.com/lindawangg/COVID-Net. In Table 2 we compare the classification metrics obtained with COVID-Net and our ResNet-18 and DenseNet-121 models: all of the models have been trained using COVID-ChestXRay, and tested on both CORDA and COVID-ChestXRay. In line with the discussion above we can note that all of the three models yield surprising results when the same dataset is used for training and testing. The performance of COVID-Net on the COVID-ChestXRay test-set (the same dataset used at training time) is very high (BA of 0.85 and DOR of 36.0) while it drops significantly when tested on CORDA, where BA is 0.55 only and DOR is 6.68. This drop can be explained by looking at the sensitivity and specificity values: it is evident that the model classifies as COVID—almost all the data. A similar behavior can also be observed in the ResNet-18 and DenseNet-121 models: the observed performance apparently is extremely high (since that the BA on the test-set reaches 1.0 for ResNet-18), and similar numbers are also claimed in the other works on ResNet-like architectures [16,17,18]. However, testing on CORDA reveals that deep models likely learn some hidden biases in COVID-ChestXRay and tend to misclassify COVID—samples as COVID+ (given that the specificity is here 0.20 for ResNet-18 and even 0.07 for DenseNet-121, having a similar phenomenon like what observed in Section 4.3). Despite some claim of having a deep model properly designed to extract the COVID feature [19], the currently low data availability limits the possibilities for the deep learning to succeed in this task. Certainly, having a pre-trained encoder to extract features from radiographic images is the most promising direction to move through.

## 5. Conclusions

One of the very recent challenges for both clinical and AI community is to use deep learning to discriminate COVID from cheap and widespread CXR. Some recent works [16,17,18,19] highlighted the possibility of successfully tackling this problem, despite the currently small quantity of publicly available data. In this work we have highlighted many obstacles towards a successful training of a deep model. Removing known biases like medical devices or textual information in the radiography and providing the deep model information strictly related to the lung content is the first practice necessary to remove some biases. Also having larger and more heterogeneous datasets could help in removing more non-trivial biases, like different settings for the acquisition machines, age, gender and ethnicity-related biases. Unfortunately, given the complexity only when the available data will scale-up by at least a factor two, or even more. Currently, the limited quantity of available data, prevents the use of large models: indeed, training smaller models is a safer choice since they are less prone to over-fit data. Very large models like DenseNet-121, if not properly regularized, tend to memorize the whole dataset with negative effects on the generalization capability.

The ongoing collection and sharing of large amount of CXR data is the only way to further investigate if promising CNN results can aid in the fight against COVID pandemic.

## Figures and Tables

**Figure 1 ijerph-17-06933-f001:**
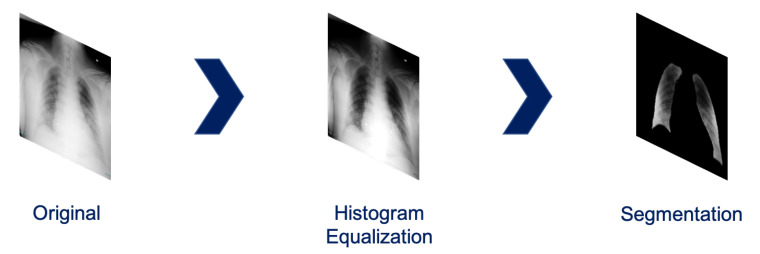
CXR pre-processing steps proposed.

**Figure 2 ijerph-17-06933-f002:**
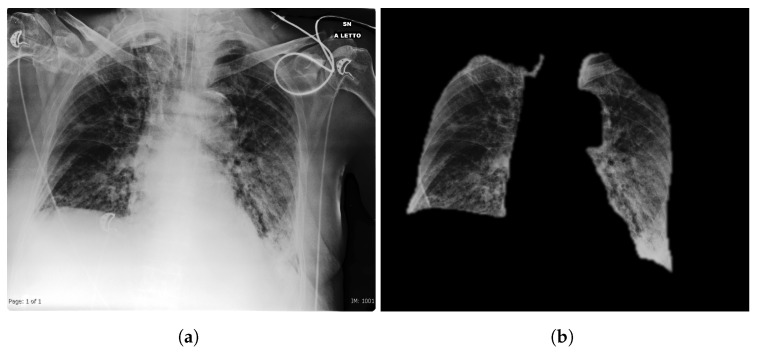
Original image (**a**) and extracted lung segmented image (**b**). Many possible bias sources like all the writings and medical equipment is naturally removed.

**Figure 3 ijerph-17-06933-f003:**
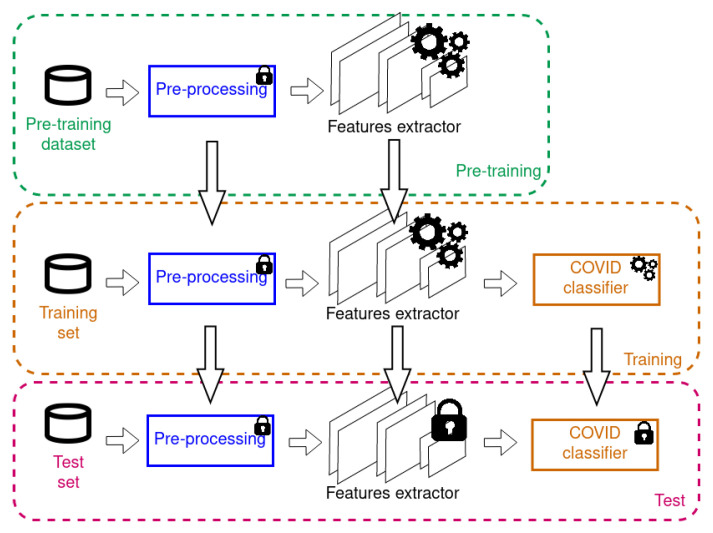
Summary of the training strategy. The feature extractor is (optionally) pre-trained on CXR pathology datasets and then fine-tuned on the COVID datasets. The presence of gears involves training/fine-tuning for the specific part, while the lock implies that part is not modified.

**Figure 4 ijerph-17-06933-f004:**
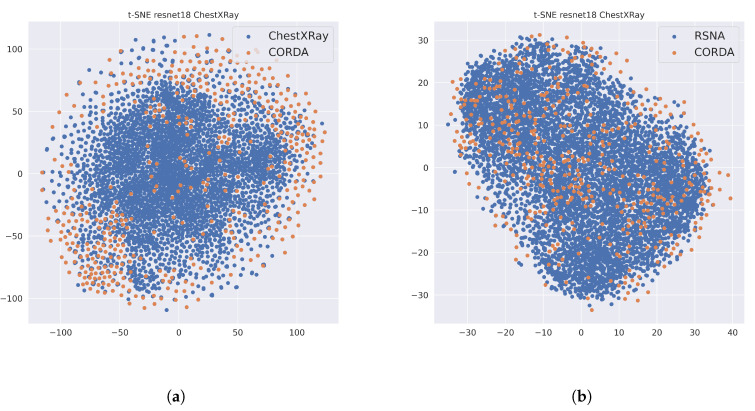
t-SNE on ChestXRay trained encoder. (**a**) shows ChestXRay data vs. CORDA data, (**b**) instead shows RSNA vs. CORDA.

**Table 1 ijerph-17-06933-t001:** Composition of datasets used for training and test are shown in the rows, with details on number of images taken from different sources in the columns; the COVID-positive samples are indicated as “+” while the negative ones with “—”. For better readability, each data source is identified by a letter: CORDA (A), ChestXRay (B), RSNA (C) and COVID-ChestXRay (D).

COMPOSED DATASET		ORIGINAL DATASETS									
		A		C		B		D		TOTAL	
		+	—	+	—	+	—	+	—	+	—
**A**	train	126	105	-	-	-	-	-	-	126	105
	test	90	45	-	-	-	-	-	-	90	45
**AB**	train	207	105	-	-	-	102	-	-	207	207
	test	90	45	-	-	-	45	-	-	90	90
**AC**	train	207	105	-	102	-	-	-	-	207	207
	test	90	45	-	45	-	-	-	-	90	90
**AD**	train	116	105	-	-	-	-	49	24	165	129
	test	90	45	-	-	-	-	10	5	100	50
**D**	train	-	-	-	-	-	-	98	24	98	24
	test	-	-	-	-	-	-	10	5	10	5

**Table 2 ijerph-17-06933-t002:** Summary for some results obtained over a number of architectures trained on various combinations of datasets. Dataset naming follows Table 1.

Architecture	Pre-Trained Encoder	Training Dataset	Test Dataset	Sensitivity	Specificity	BA	AUC	DOR
ResNet-18	none	AB	AB	0.88	0.94	0.91	0.97	112.93
	none	AB	AD	0.87	0.20	0.54	0.60	1.67
	none	A	A	0.56	0.58	0.57	0.59	1.71
	B	AB	AB	0.88	0.94	0.91	0.97	122.67
	B	AB	A	0.88	0.24	0.56	0.66	2.32
	C	A	A	0.54	0.80	0.67	0.72	4.78
	C	AB	AB	0.82	0.95	0.89	0.97	89.14
	C	AB	A	0.82	0.38	0.60	0.63	2.81
	B	D	A	0.91	0.20	0.56	0.61	2.56
	B	D	D	1.00	1.00	1.00	1.00	*∞*
ResNet-50	B	D	AB	0.98	0.72	0.85	0.90	112.57
	B	D	AC	0.98	0.11	0.55	0.61	5.59
	B	D	AD	0.98	0.20	0.59	0.65	12.25
COVID-Net	B	D	A	0.12	0.98	0.55	0.55	6.68
	B	D	D	0.90	0.80	0.85	0.85	36.00
DenseNet-121	B	D	A	0.99	0.07	0.53	0.61	6.36
	B	D	D	1.00	0.80	0.90	0.98	*∞*

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
