# Peer review of "Unveiling COVID-19 from CHEST X-Ray with Deep Learning: A Hurdles Race with Small Data"

_ijerph, 2020, doi:10.3390/ijerph17186933_

Round 1

Reviewer 1 Report

The authors evaluated different combinations of usage of existing datasets and transferred learning of standard CNN models. The results showed that small bias could have huge impacts on the performance's significance because of small and local datasets. The authors pointed out that researchers need to keep collecting and sharing CXR data for using CNN results against the current pandemic.

The experiments are well designed, and the results are reliable. The only confusing part is in the 2.1 datasets section, the author listed six datasets but wrote three different datasets will be used. In the table 1, the author took data from four datasets and composed five datasets for analysis. Authors need to make the part clearer.

Author Response

Point 1: Thank you for raising the attention on the use of the datasets. All the six datasets described have been used in our experiments: among those, four provide label for classification tasks (CORDA, ChestXray, RSNA and COVID-ChestXray) and two (Montgomery County X-ray Set and Shenzhen Hospital X-ray Set) provide ground-truth masks and are used to train the segmentation of the lungs. This point has been clarified in the revised manuscript in Section 2.1.

Reviewer 2 Report

The paper is of great importance and thoroughly documents how the identification lung lesions due to COVID-19 can be identified with deep learning algorithms.

I have suggestions on improving the accessibility of the paper:

  1. The description of training in Sec. 2.3 should be accompanied with a graph for better understanding of the algorithm design.
  2. The tables 2-4 should be relegated to the appendix and replaced by more legible graphs in the main text. Otherwise, the reader has to put in a lot of work to reach the conclusions.
  3. The conclusion of the paper should provide also recommendation for the future use of algorithms and the suitable direction of further development in that regard, not just related to data.

Author Response

Point 1: Thank you for the suggestion. A graph (Fig.3) has been included to give a general overview of the algorithm design.

Point 2: Following the recommendation, we have moved Tables 2-4 in the appendix (revised Tables A1-A4) and replaced those with the new Table 2 that reports only the main results that are useful for discussion in Section 4.

Point 3: We have included recommendations on future directions for algorithm usage, according also to some insights provided by the medical personnel.

Reviewer 3 Report

The authors present a deep learning algorithm for automated diagnosis of COVID-19 from chest radiography using many public datasets and their own datased. They also comment on the methods and difficulties of performing this task. The paper is interesting, as not much deep learning algorithms have been trained on radiographies which is of interest as it is a widely used imaging methodology. The manuscript suffers from some major flaws that need to be addressed before pubblication. First, the paper looks like a mix between a research paper and a review, and this is very confounding because it feels like the results are not properly presented and discussed but are lost among many recommendations and observations. These are often in the sections where the reader would expect the materials and methods used in the research, or the results.

Moreover, many of these recommendations (e.g. deal with imbalanced datasets, ) are very obvious and can be fond in any review on deep learning in any context (e.g. oncology). They are ok, but could be condensed.

The paper should be revised in order to be more specific on the application which is the goal of the study, that is COVID diagnosis.

Finally, the language of the manuscript should be improved, possibly with collaboration from a native english speaking coauthor.

Page 6 line 196: "After data have been pre-processed, a deep model will be trained. Towards this end, the following choices have been taken: " why the future tense was used? the past tense should be used consistently through all the paper

Line 213: "Balancing the training data is yet another extremely important issue to be considered." It is not clear what are the authors referring to here. Do they mean datasets with lower number of positives than negatives, or confounding factors? Moreover, authors should give some indications on how tackle this issue besides removing patients (!). Are there data augmentation algorithms used to oversample the data?

The last point in this page can be omitted: three lines to say that there are many other issues!Besides: is relly validation used to tune the model?

The results section is written as matherial and methods, and should be written in order to present the results. 

In Table 2, it seems that the validation and training dataset are allowed to be the same. It shouldn't happen, but I also wonder why in this case the accuracy is not much higher than when training and validation sets are different.

t-SNE is never introduced in the paper!

which data augmentation techniques e.g. image reflection, translation, scaling, have been used?

In section 4.5, I recommend to state clearly which is the best model obtained and that looks promising for future clinical application.

The conclusion is too generical: it just says that obtaining a model is difficult, which would fit in the discussion maybe. Rewrite the conclusion in order to summarize the main findings of the manuscript.

Author Response

Point 1: Following your suggestion, we have proposed, in the main text, the most relevant results only: all the performed experiments have been moved to Appendix. Towards this end, we have also re-arranged the section “Results” according to your other comments.

Point 2: We agree with the reviewer, and we have synthesized and referenced some more common recommendations (like in page 6, about the dataset imbalance). 

Point 3: We thank the reviewer for raising its concerns about English tenses through all the paper, which have been double-checked and corrected according to suggestions.

Point 4: Training set and validation set are never allowed to be the same, through all the paper. Each of the discussed datasets are divided into a training and a validation set: testing results can be reported between any dataset (when the data are consistent, like in our case). We have further emphasized this point in Sec.2.3

Point 5: t-SNE is a standard nonlinear dimensionality reduction algorithm used in machine learning. We understand that it might not be obvious for the audience this paper addresses to and we have introduced and described it in Sec. 4.3

Point 6: Currently, when training is performed using all the recommendations which are proposed through the paper, a small model like ResNet-18 is sufficient to produce the best results. However, when big data will be available, then a performance gap between ResNet-18 and more sophisticated architectures, like DenseNet-121, will favor the more complex of the two. However, we would require an exponentially larger number of CXRs than the one currently available. This discussion has been included in the paper.

Point 7: Conclusions have been revised by better emphasizing the most important message of the work, i.e. the obstacles that must be considered when detecting COVID from CXRs given the current state of the art.